# A method for quantitatively separating the piezoelectric component from the as-received "Piezoelectric" signal

Chaojie Chen [1], Shilong Zhao[1], Caofeng Pan [2], Yunlong Zi [3], Fangcheng Wang[1], Cheng Yang [1✉] & Zhong Lin Wang [2,4✉]

Polymer-based piezoelectric devices are promising for developing future wearable force sensors, nanogenerators, and implantable electronics, etc. The electric signals generated by them are often assumed as solely coming from the piezoelectric effect. However, triboelectric signals originated from contact electrification between the piezoelectric devices and the contacted objects can produce non-negligible interfacial electron transfer, which is often combined with the piezoelectric signal to give a triboelectric-piezoelectric hybrid output, leading to an exaggerated measured "piezoelectric" signal. Herein, a simple and effective method is proposed for quantitatively identifying and extracting the piezoelectric charge from the hybrid signal. The triboelectric and piezoelectric parts in the hybrid signal generated by a poly(vinylidene fluoride)-based device are clearly differentiated, and their force and charge characteristics in the time domain are identified. This work presents an effective method to elucidate the true piezoelectric performance in practical measurement, which is crucial for evaluating piezoelectric materials fairly and correctly.

[1] Institute of Materials Research, Tsinghua Shenzhen International Graduate School, Tsinghua University, Shenzhen, PR China. [2] CAS Center for Excellence in Nanoscience, Beijing Key Laboratory of Micro-nano Energy and Sensor, Beijing Institute of Nanoenergy and Nanosystems, Chinese Academy of Sciences, Beijing, PR China. [3] Department of Mechanical and Automation Engineering, The Chinese University of Hong Kong Shatin, N.T., Hong Kong, China. [4] School of Materials Science and Engineering, Georgia Institute of Technology, Atlanta, GA, USA. ✉email: yang.cheng@sz.tsinghua.edu.cn; zhong.wang@mse.gatech.edu

Mechanical energy sources are almost everywhere in the surrounding environment, e.g. wind, wave, and human motion. Harvesting these energies from the environment, so as to provide electricity to smart terminals and other equipment, has become an attractive and competitive way-out in the era of Internet-of-Things[1–6]. In recent years, piezoelectric nanogenerators (PENGs) and triboelectric nanogenerators (TENGs) as new kinds of energy harvesters have attracted much attention, especially in the field of wearable devices[7–13]. The polymer-based PENGs and TENGs are even more attractive because they are flexible, light-weighted, and easy-to-process, which can be used to scavenge mechanical energy at multi-scales especially from human motion[14–19].

Contact electrification (CE) as a universal electron transfer phenomenon that can occur between any material in different physical states (solid, liquid, and gas)[20]. It is often coupled with electrostatic induction to generate electric signals in the process of repeated contact and separation of two different materials, which is the working mechanism of TENGs. Recently, Sutka et al.[21] confirmed that CE can take place in piezoelectric devices during a variety of motions, including interfacial shear friction, nanofiller slippage, pre-existing static charge and contact friction. Simultaneously, the existence of triboelectric signals generated by the contacted objects and piezoelectric devices can be strong and shouldn't be ignored, especially under repeated pressing. However, some previous works ignored this kind of triboelectric signal and regarded the output from piezoelectric devices as the sole piezoelectric signal, leading to exaggerated performance. The lack of a reliable method to differentiate piezoelectric signals and triboelectric signals obstructs us reveal the influence of triboelectric signals and evaluate the true contribution of the piezoelectric effect, which results in incorrect evaluation on piezoelectric performance. More specifically, because TENG and PENG have the same output including voltage, current, and charge, traditional analysis on electric signals cannot identify their respective part in a hybrid output. Therefore, developing a method to identify and extract the piezoelectric component from a "piezoelectric" signal is particularly important for quantitively evaluating the performance of piezoelectric materials.

In the present work, we analyzed the CE process that happened between loading objects and the encapsulating materials, and developed an effective method to identify the piezoelectric part from a hybrid triboelectric-piezoelectric output in a sandwich-structured poly(vinylidene fluoride) (PVDF)-based piezoelectric device system. Based on our experimental observation, the triboelectric signals can take a large portion, which should not be ignored. Here we investigated the force-time curve of the piezoelectric process as a strong supplement rather than solely analyzing the electric signals, which is the source of the electric signals. By comparing these two curves simultaneously, it can be found that before and after the device-object contact, the recorded electric signals belong to the triboelectric contribution, and the electric signals acquired when they are contacted can be ascribed to the piezoelectric effect. Then the piezoelectric charge transfer was quantitatively extracted from the hybrid output and the effective piezoelectric coefficient ($d_{33}$) was calculated, such results were consistent with those measured by other methods, which are elucidated as follows. In such way, we can conveniently separate the combined signals clearly, and the accuracy is only limited by both force-time and charge-time resolutions.

## Results

### Piezoelectric-triboelectric hybrid output in piezoelectric devices

Figure 1a shows the schematic of the PVDF-based piezoelectric device in pressure sensing. CE can occur between the human finger and the piezoelectric device—the human finger would have positive charges and the protective layer of the device would have negative charges. As a result, a single-electrode triboelectric nanogenerator (SE-TENG) will form when the human finger is approaching the piezoelectric device again. Its working mechanism is illustrated in Fig. 1b, including four typical stages, such as contacted, separating, separated, and contacting. Among these stages, the triboelectric signals are generated in the stages of contacting and separating. On the other hand, the PENG will be formed when the human finger is pressing the piezoelectric devices (right in Fig. 1a), and its working mechanism is clearly illustrated in Fig. 1c. The piezoelectric signals would be generated in the stages of pressing and releasing. Therefore, the SE-TENG and PENG exist during the practical application of piezoelectric devices, resulting in the triboelectric-piezoelectric hybrid output.

### Verification of the existence of the SE-TENG system in piezoelectric devices

In order to verify the existence of the SE-TENG system during the measurement of piezoelectric devices, here we replaced the PVDF film with a non-piezoelectric material polyimide (PI) for investigation. This means that the interference from piezoelectric signals was eliminated and thus only triboelectric signals will exist during the measurement. The testing platform is illustrated in the experimental section and Supplementary Fig. 1. During the measurement, two different wiring methods were adapted including forward connection and reverse connection so as to verify whether the electric signals were generated by the testing system itself. More specifically, if there were not any interference from other signals, the electric signals generated by the testing system would symmetrically flip with constant amplitude when the connection direction was switched[22,23].

Figure 2a shows a schematic diagram of measuring electric signals between the front electrode and back electrode of the PI-based device. Open-circuit voltage (Fig. 2d) and transferred charge (Fig. 2g) curves show significant electric signals in the forward connection, while electric signals obtained in the reverse connection are very low. Thus the electric signals in the forward and reverse connection are not symmetrical along the x-axis, suggesting they are not generated by the testing system composed of the front and back electrode. This is because, in the forward connection, the Al plate, Kapton layer, and the front electrode form a SE-TENG, then triboelectric signals can be detected when connecting the signal probe with the front electrode (Supplementary Fig. 2a). By contrast, positive charges in the front electrode would flow to the ground in the reverse connection and thus weaker electric signals can be detected in the back electrode (Supplementary Fig. 2b), indicating that the front electrode acted as a shielding layer.

To verify that CE did occur between the Al plate and the Kapton layer, two kinds of contact-separation triboelectric nanogenerators (CE-TENGs) were tested (CS-TENG-1 in Fig. 2b and CS-TENG-2 in Fig. 2c). For CS-TENG-1, it is composed of Al plate, Kapton layer, and front electrode. The voltage signals (Fig. 2e) generated by CS-TENG-1 in the forward connection and in the reverse connection are symmetrical along the x-axis. More specifically, the signs of voltage signals obtained in two connections are opposite and the amplitudes of voltage signals are constant. The charge signals (Fig. 2h) of CS-TENG-1 have the same characteristics with voltage signals. For CS-TENG-2, it is composed of Al plate, Kapton layer, front electrode, PI film, and back electrode. Similarly, the voltage signals (Fig. 2f) of CS-TENG-2 obtained in the forward connection and in the reverse connection are symmetrical along the x-axis; the charge signals (Fig. 2i) of CS-TENG-2 obtained in the forward connection and in the reverse connection are symmetrical along the x-axis too.

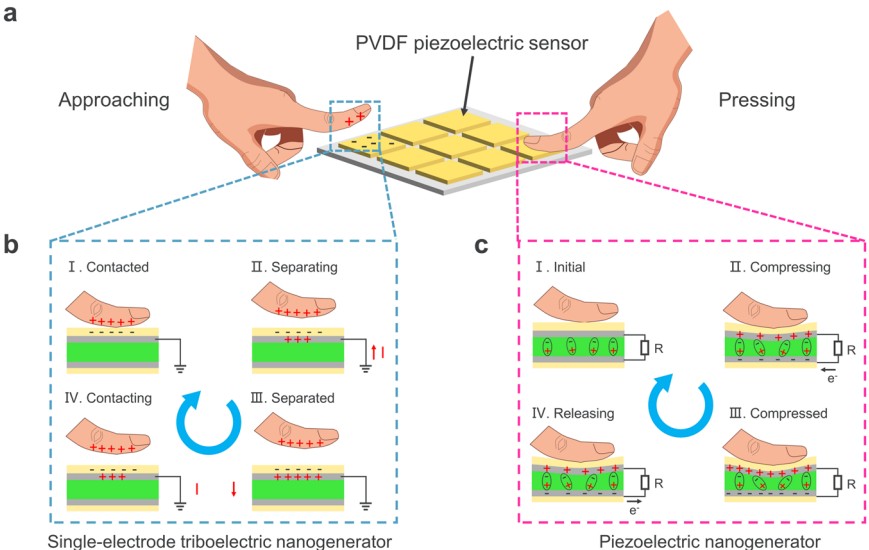

**Fig. 1 Schematic of the existence of both SE-TENG and PENG during practical application of piezoelectric devices, but the time at which the signal is generated has distinct differences. a** The human finger has positive charges and the surface of a piezoelectric pixel has negative charges (left). The piezoelectric material would experience elastic deformation when a human finger is pressing it and the piezoelectric effect would occur (right). **b** A SE-TENG is formed when using the positive finger touches the negative piezoelectric pixel again. The working mechanism of SE-TENG contains four stages including contacted, separating, separated, and contacting, the triboelectric signals will be generated in the stages of separating and contacting. **c** The working mechanism of PENG contains four stages including initial, compressing, compressed, and releasing. Piezoelectric signals will be generated in the stages of pressing and releasing.

The above results indicate that the electric signals are generated by CS-TENGs where the Al plate and the Kapton layer constitute the triboelectric pairs, and their working mechanism is illustrated in Supplementary Fig. 3. Furthermore, three compressing materials with different electron affinities were introduced including fluorinated ethylene propylene (FEP), PI, and Al, to compress the Kapton layer. Compared with Kapton, FEP tends to gain electrons, while Al tends to lose electrons. As a result, the directions of charge curves are opposite between FEP-Kapton and Al-Kapton (Supplementary Fig. 4a, c). The PI-Kapton triboelectric series can also induce charge transfer (Supplementary Fig. 4b), but the value is small ($-0.3$ nC) compared with FEP-Kapton (1 nC) and Al-Kapton ($-1.5$ nC) in Supplementary Fig. 4d, indicating that the interference from CE can be reduced using the same compressing material as the encapsulation material. Therefore, the SE-TENG composed of compressing material and protective layer was confirmed by the experiments, which is consistent with the theoretical analysis.

It is worth noting that the peak voltage (1.5 V) and charge transfer ($-1.1$ nC) generated by SE-TENG are too large compared with polymer-based piezoelectric materials (such as PVDF, the peak voltage of them usually is several volts[24,25]). Simultaneously, using a finger to tap the PI-based device can generate electric signals up to ~7 V and ~$-4$ nC (Supplementary Fig. 5), which is larger than sole piezoelectric charges generated under different forces (Supplementary Fig. 11a, b). As a result, these triboelectric charges should not be ignored.

**Involving force signals to differentiate the triboelectric signals and piezoelectric signals**. It is essential to differentiate the triboelectric signals and piezoelectric signals before quantitively obtaining the piezoelectric charge transfer and evaluating the piezoelectric performance. Here we developed an effective method that can find the sources of the resulting electric signals by analyzing the force loading process of piezoelectric devices.

And this simple method was demonstrated by successfully identifying the sole triboelectric signals and piezoelectric signals.

The sole triboelectric signals are generated by the PI-based device (Fig. 3a). The sole piezoelectric signals are generated by the PVDF-based device with a conductive shielding layer (Fig. 3b). In addition, the role of shielding layer is illustrated in Supplementary Fig. 6; the PI-based device with a shielding layer (Supplementary Fig. 6a) cannot generate any electric signals (Supplementary Fig. 6c, e), while the PVDF-based device with a shielding layer (Supplementary Fig. 6b) can generate symmetrical pure piezoelectric signals in the forward and reverse directions (Supplementary Fig. 6d, f).

According to the force signal (Fig. 3i), the sole triboelectric signal generation process can be divided into three stages including contacting, contacted, and separating. In the contacting stage, the Al plate is contacting with the PI-based device. At the same time, the short-circuit current $I_{SC}$ (Fig. 3c) and charge curve (Fig. 3g) fall gradually, while the open-circuit voltage $V_{OC}$ signal (Fig. 3e) rises slowly. When Al plate and the PI-based device are contacted with each other, the electrostatic balance between them is formed; the $I_{SC}$ signal backs to the baseline quickly, while the $V_{OC}$ and transferred charge signals remain unchanged. In the separating stage, the Al plate is separating from the PI-based device, which breaks the electrostatic balance that existed before, resulting in the signal rise of the $I_{SC}$. Besides, the $V_{OC}$ and transferred charge signals go back to the baseline gradually. Differently, the $I_{SC}$ can reach to peak when the Al plate and the PI-based device are separated, and then falls to the baseline. Overall, there is a phase difference between the sole triboelectric signal and force signal.

By contrast, according to the force signal (Fig. 3j), the sole piezoelectric signal only occurs in the contacted stage (light green area) and its generation process can be divided into three stages including compressing, compressed, and releasing. In the compressing stage, the PVDF film experiences a repaid elastic deformation and thus the $I_{SC}$ (Fig. 3d) and charge signals

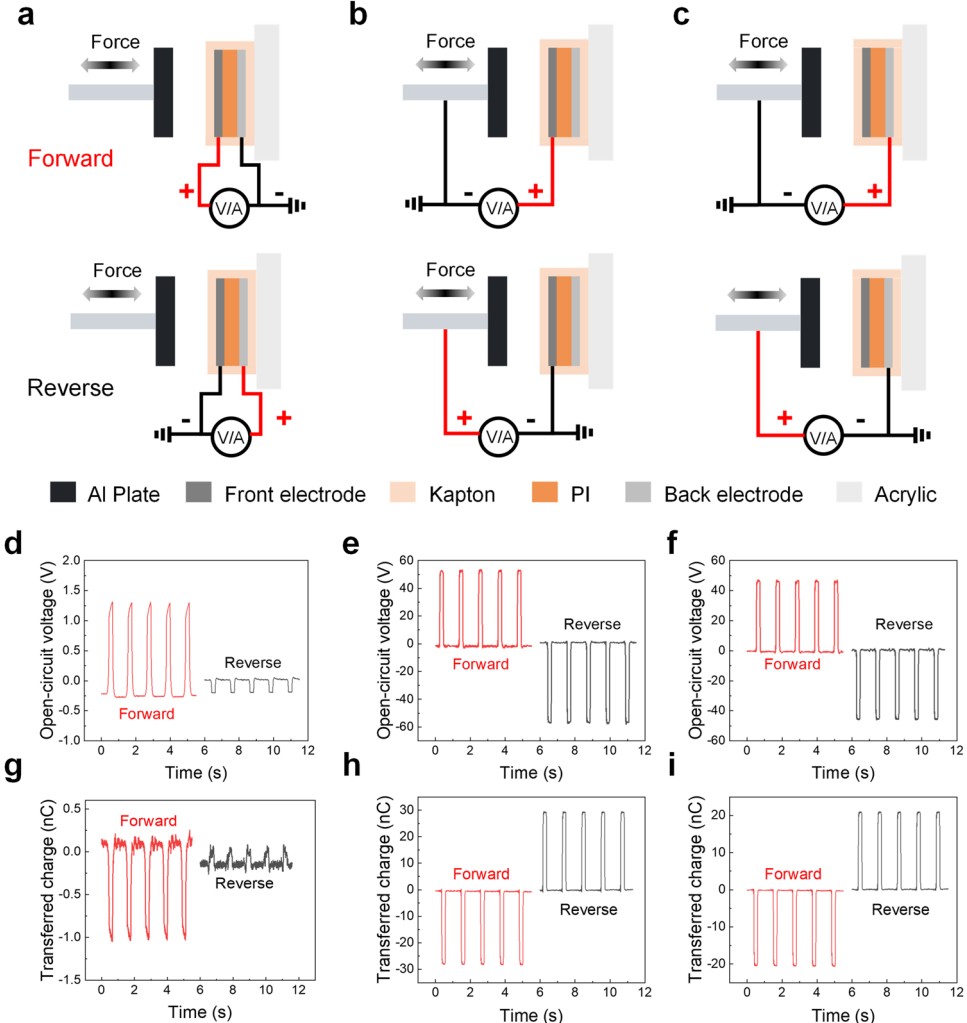

**Fig. 2 Verifying the existence of SE-TENG in a compression test. a** Schematic diagrams of the forward connection and reverse connection when measuring the electric signals between the front electrode and the back electrode of the PI-based device. The corresponding open-circuit voltage and transferred charge results are shown in **d** and **g**, respectively. **b** Schematic diagrams of the forward connection and reverse connection when measuring the electric signals between the front electrode and the Al plate. The corresponding open-circuit voltage and transferred charge results are shown in **e** and **h**, respectively. **c** Schematic diagrams of the forward connection and reverse connection when measuring the electric signals between the back electrode and the Al plate. The corresponding open-circuit voltage and transferred charge results are shown in **f** and **i**, respectively.

(Fig. 3h) reach to peak dramatically, while the $V_{OC}$ (Fig. 3f) falls to the lowest point. In the compressed stage, the elastic deformation of PVDF film reaches the maximum value, indicating that the piezoelectric effect would not be activated further. As a result, the $I_{SC}$ cannot increase further and falls to the baseline; the $V_{OC}$ and charge signal remain unchanged. In the releasing stage, the elastic deformation of PVDF film is releasing rapidly. The $V_{OC}$ and charge signals go back to the baseline, while the $I_{SC}$ decreases to the lowest point when the elastic deformation is released and then rises back to the baseline. More importantly, the $V_{OC}$ and the charge signals have the same trend with the force curve, and there are no phase differences existed. Therefore, analyzing the force signal of devices is an effective method for differentiating triboelectric signals and piezoelectric signals.

Comparing these two kinds of sole signals, the rising edge of the piezoelectric charge signal (Fig. 3h) is non-smooth that is consistent with the force signal (Fig. 3j); while triboelectric signals do not show this characteristic (Fig. 3e, i). Besides, triboelectric signals have a longer response time (115.61 ms) than piezoelectric signals (32.81 ms). This is because, although triboelectric signals and piezoelectric signals are the results of the displacement current, their actual displacement distances are different[26]. Triboelectric signals can be generated over longer displacement (up to millimeter scale[27]) because of the electrostatic induction, resulting in a longer response time. The limited deformation of PVDF film causes shorter displacement that enables piezoelectric signals to reach peak more quickly. Triboelectric signals are mainly dependent on contact area, while piezoelectric signals are dependent on force. Consequently, they both increased from 40 N to 90 N due to the increased contact area and forces (Supplementary Fig. 7a, b), but the piezoelectric signals show good linearity compared with triboelectric signals (Supplementary Fig. 7c). These two kinds of signals show frequency-independent behavior at low frequency from 0.5 Hz to 2.5 Hz (Supplementary Fig. 8a, b) due to the fixed force.

**Identifying the piezoelectric signal in "piezoelectric" output.** Above results and discussion demonstrated that the effectiveness of differentiating sole triboelectric signals and sole piezoelectric signals by analyzing the force loading curve. Now we can identify

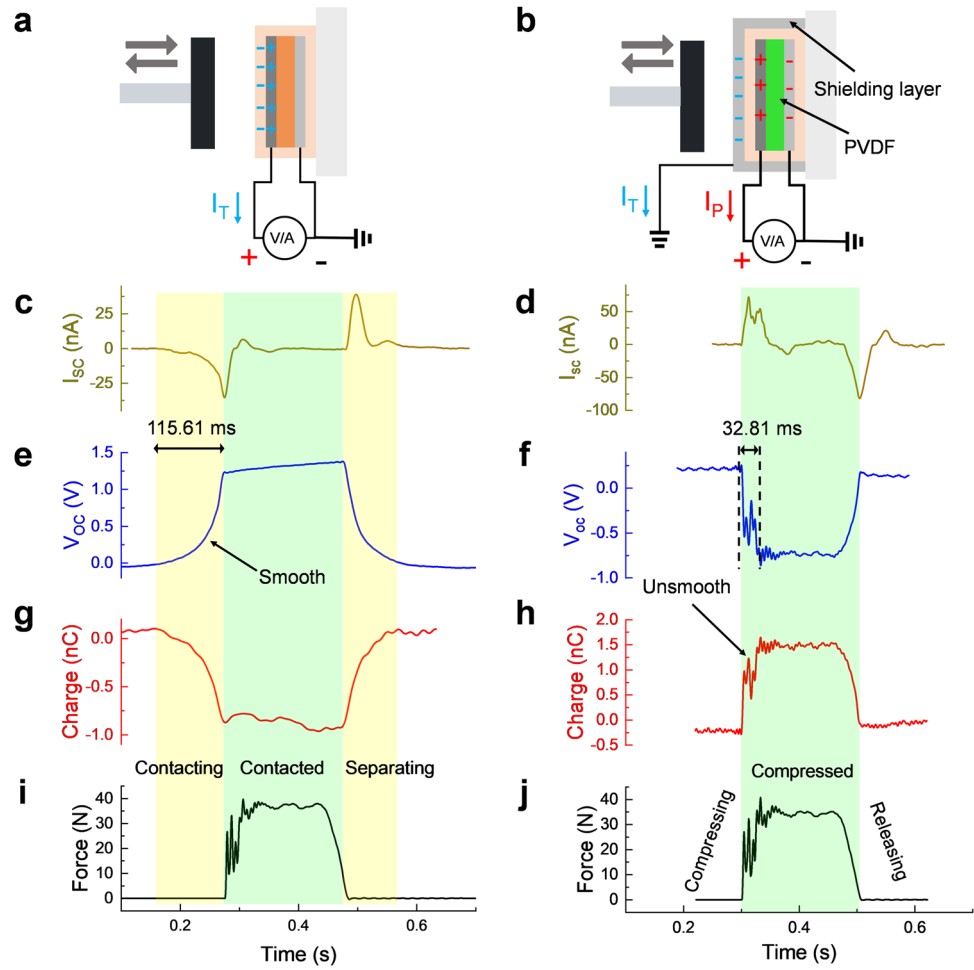

**Fig. 3 Using loading force signal to differentiate the sole triboelectric signals and the sole piezoelectric signals. a** Schematic diagram of measuring sole triboelectric signals from the PI-based device. Sole triboelectric signals including **c** short-circuit current, **e** open-circuit voltage, and **g** transferred charge are obtained under a certain **i** force signal, respectively. **b** Schematic diagram of measuring sole piezoelectric signals from PVDF-based device with a shielding layer. The conductive shielding layer is attached on the device's surface, which can guide the induced triboelectric charges ($I_T$) flow to the ground and only piezoelectric signals ($I_P$) can be obtained in the front electrode. Sole piezoelectric signals including **d** short-circuit current, **f** open-circuit voltage, and **h** transferred charge are obtained under a certain **j** force signal, respectively.

the piezoelectric signals from the "piezoelectric" output. Two measurements were carried out by taking into account the direction difference between triboelectric signals and piezoelectric signals. This is because, the direction of triboelectric signals is determined by the triboelectric series and keeps constant[28], and that of piezoelectric signals depends on the polarization direction of signal acquisition side.

Figure 4a is a schematic of measuring electric signals from the negative polarization side of the PVDF film. The signal generation process of the hybrid output can be divided into six parts, including contacting, contacted, compressing, releasing, released, and separating (Fig. 4b). Fig. 4c shows the sole triboelectric signals (Supplementary Movie 1) and corresponding force signal of the PI-based device; Fig. 4d shows the sole piezoelectric signals (Supplementary Movie 2) obtained in the negative polarization side of PVDF-based device with a shielding layer and its corresponding force signal; Fig. 4e shows the triboelectric-piezoelectric hybrid signals (Supplementary Movie 3) generated by the PVDF-based device. By manipulating the direction of the piezoelectric signals to be opposite with the triboelectric signals, the turning points (II and V) of electric signals can be observed in Fig. 4e. The existence of turning points

not only differentiates the piezoelectric part (light green area) and triboelectric part (light yellow area) in the hybrid output, but also proves the difference between piezoelectric signals and triboelectric signals in the time domain which is discussed before. Therefore, the hybrid output can simply be regarded as the sum of the sole triboelectric signals and the piezoelectric signals.

However, if flipping the piezoelectric device and collect the hybrid output from the positive polarization side of the PVDF film (Supplementary Fig. 9a), things are different (Supplementary Fig. 9b). The hybrid output (Supplementary Fig. 9e, Movie 4) can still be regarded as the sum of sole triboelectric signals (Supplementary Fig. 9c) and sole piezoelectric signals (Supplementary Fig. 9d, Movie 5), while there are no turning points in the stages of contacted and released due to the direction of piezoelectric signals is the same as triboelectric signals.

Overall, by investigating the force loading signal of the devices, it's very convenient to determine whether the resulting voltage signal is the triboelectric signal, the piezoelectric signal, or hybrid signal from the perspective of the time domain, while traditional electrical signal analytical methods cannot do.

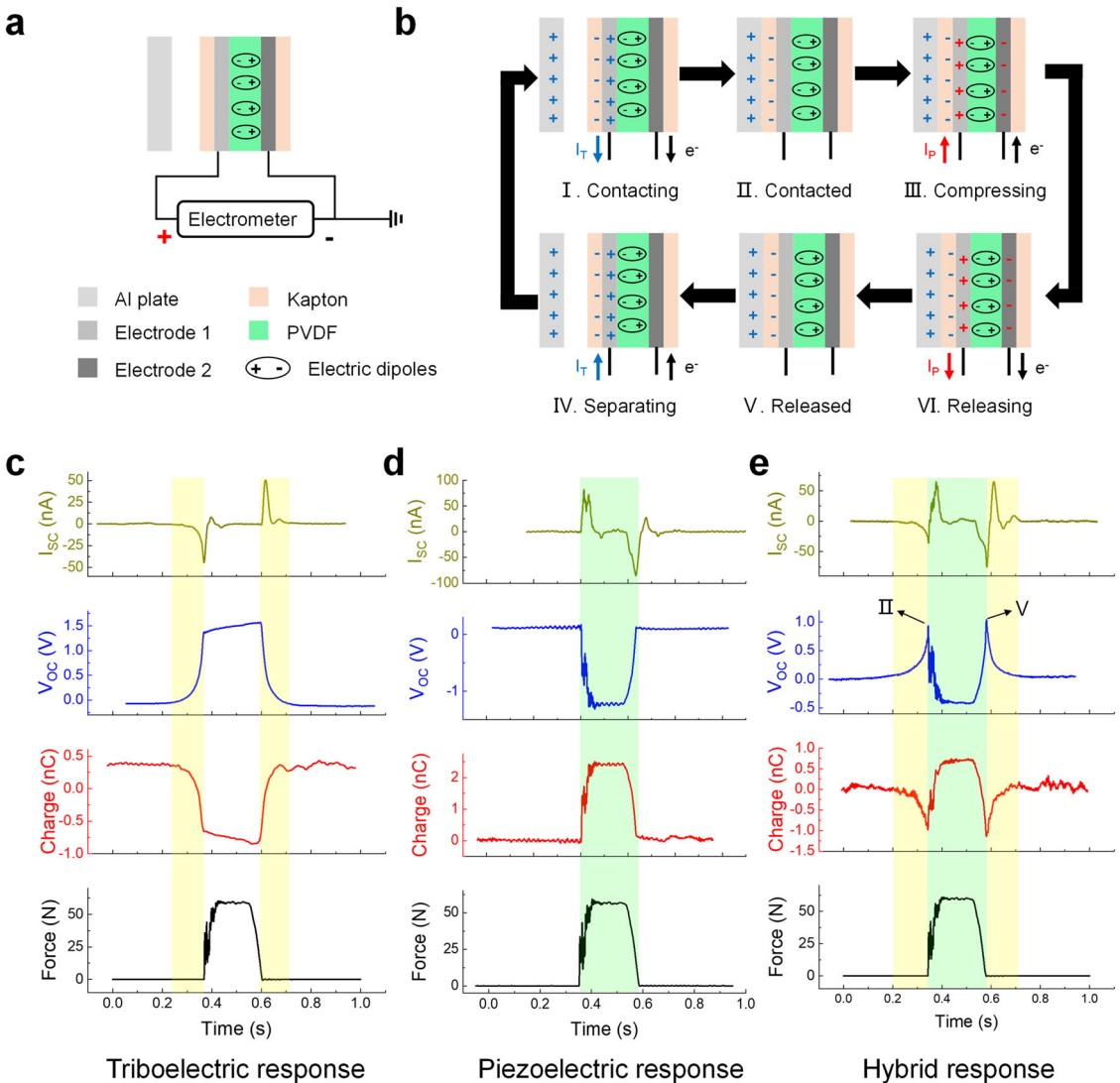

**Fig. 4 The triboelectric-piezoelectric hybrid output generated by the PVDF-based device with the negative polarization direction. a** Schematic of measuring the triboelectric-piezoelectric hybrid output in the negative polarization direction of the PVDF film. **b** The signal generation process of hybrid output contains six stages including contacting, contacted, compressing, releasing, released, and separating stages. **c** The sole triboelectric signals ($I_{SC}$, $V_{OC}$, and charge) are generated by the PI-based device at 60 N. Sole triboelectric signals are generated in the stages of contacting and separating (light yellow area). **d** The sole piezoelectric signals ($I_{SC}$, $V_{OC}$, and charge) are generated by the PVDF-based device with shielding layer at 60 N. Sole piezoelectric signals are generated in the stages of compressing and releasing (light green area). **e** The triboelectric-piezoelectric hybrid signals ($I_{SC}$, $V_{OC}$, and charge) are generated by the PVDF-based device at 60 N. The hybrid signals exist in the stages of contacting, compressing, releasing, and separating.

**Extracting the piezoelectric charge transfer from "piezoelectric" output for piezoelectric performance evaluation.** Here, we define $Q_p$ as the piezoelectric charge transfer that induced by PVDF film at a certain force. Piezoelectric charges would accumulate with the increased force at the compressing stage and saturate in the compressed stage, finally getting released in the releasing stage. During the above stages, the saturated charge value at the end point of compressing stage (start point of compressed stage) is $Q_p$. Based on above results and discussion, triboelectric charges and piezoelectric charges are saturated in the compressed stage as well as an electrostatic balance is formed, so we proposed a compressed balance analysis (CBA) method to extract piezoelectric charge transfer from the hybrid output. Specifically, as shown in Fig. 5a (I), there is an electrostatic balance between triboelectric charges and piezoelectric charges at the compressed stage. $Q_2$ is the transferred charge in the Al plate, $-Q_1$ is then transferred charge in the Kapton layer, $q$ is the total

charge transfer in the electrode which contains both piezoelectric and triboelectric parts (Fig. 5b), $Q_p$ is the induced piezoelectric charge. Their relationship is:

$$Q_2 - Q_1 + q - Q_p = 0 \qquad (1)$$

After flipping the device (II in Fig. 5a), the electrostatic balance becomes:

$$Q_2 - Q_1 - q' + Q_p = 0 \qquad (2)$$

Where $-q'$ is the total charge transfer in the electrode (Fig. 5c). By making Eq. (1) – Eq. (2):

$$Q_p = \frac{q + q'}{2} \qquad (3)$$

As shown in Eq. (3), the total induced piezoelectric charge $Q_p$ can be calculated by substituting the values of $q$ and $q'$ into

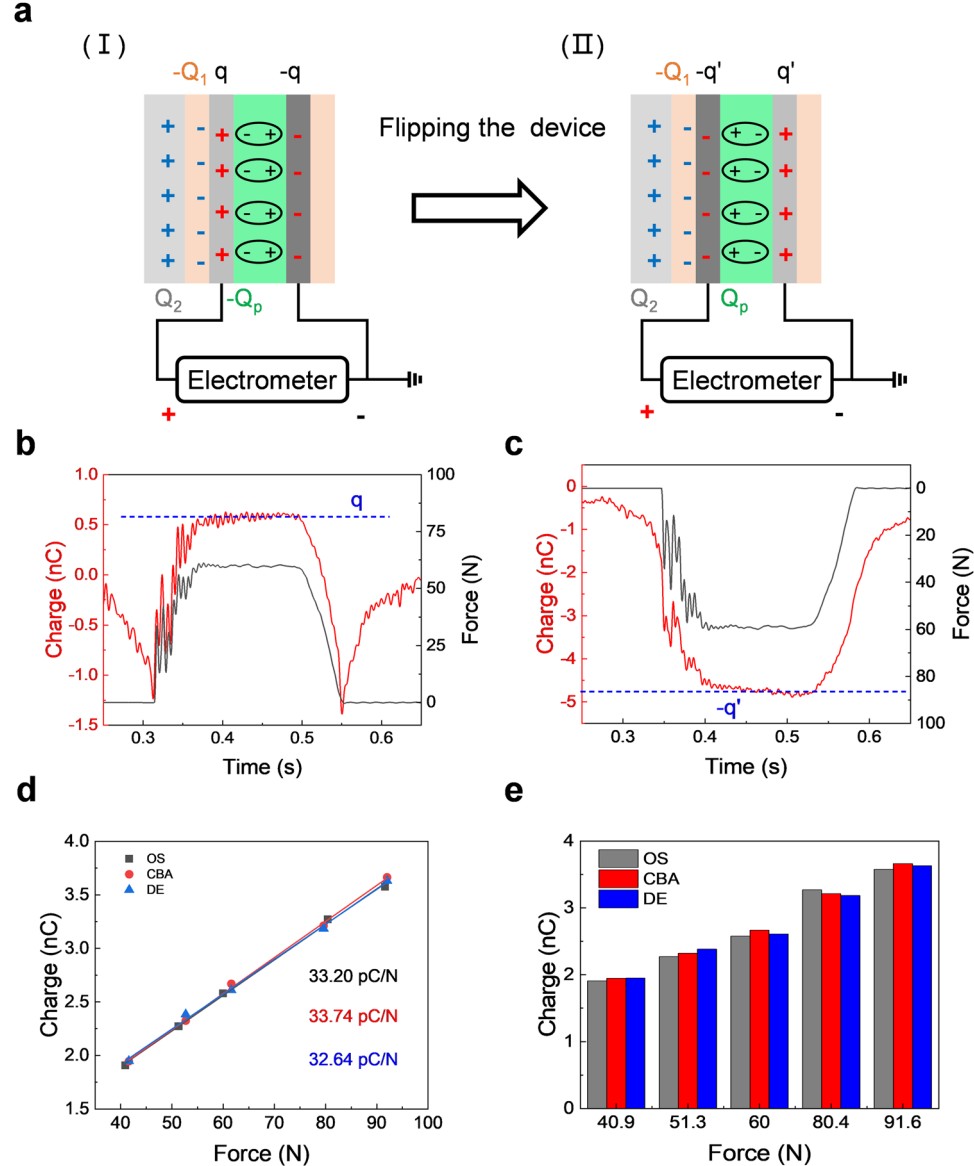

**Fig. 5 Evaluating the piezoelectric performance of PVDF film by extracting the piezoelectric charge transfer from the hybrid output. a** The electrostatic balance is achieved when the Al plate and the PVDF-based device are contacted. I represent the positive polarization, II represents the negative polarization, $Q_2$ is the total charge in the Al plate, $-Q_1$ is the total charge in the Kapton layer, $q$ is the charge transfer between electrodes in the positive polarization, $Q_p$ is the induced piezoelectric charge, $-q'$ is the charge transfer between electrodes in the negative polarization. The transferred charges between electrodes are measured from the **b** positive and **c** negative polarization sides of PVDF-based device, and the applied force is 60 N. **d** The charge–force curves of PVDF-based device are obtained by three methods including outside shielding (OS), compressed balance analysis (CBA), and directly extracting (DE). The effective $d_{33}$ of OS is 33.20 pC/N, that of CBA is 33.74 pC/N, and that of DE is 32.64 pC/N. **e** The piezoelectric charge transfer under different forces is obtained by three methods including OS, CBA, and DE. They are almost equal.

expression without considering the interference from triboelectric charges. Furthermore, the effective piezoelectric coefficient $d_{33}$ of PVDF film can be obtained by Eq. (4), where $F$ represents the applied force.

$$d_{33} = \frac{Q_p}{F} \qquad (4)$$

The transferred charge curves measured in the positive and negative polarization direction of the PVDF-based device under different forces are shown in Supplementary Fig. 10a, b, respectively. Additionally, the piezoelectric charge transfer can be extracted directly from the hybrid output by calculating the relative value in the stage of compressing (Supplementary Fig. 10c, d, and Note 1), which is called directly extracting (DE) method.

In order to verify the effectiveness of the above two methods, an outside shielding (OS) method was introduced to obtain the sole piezoelectric charge transfer (details are shown in Supplementary Fig. 11 and Note 2), which can be regarded as a benchmark because the triboelectric part is eliminated. Figure 5d shows the charge-force curves obtained by the above three methods. Importantly, the effective $d_{33}$ of CBA is 33.74 pC/N and that of DE is 32.64 pC/N, which are nearly consistent with the shielded device (33.20 pC/N). Simultaneously, the transferred piezoelectric charges calculated by these three methods in different force loading are almost equal (Fig. 5e). Furthermore, a commercial piezoelectric meter is also introduced to measure the $d_{33}$ of the PVDF film (Supplementary Fig. 12). The average $d_{33}$ measured by this instrument is 33.05 pC/N, which is consistent with the

aforementioned results, indicating that these three methods are both effective. By contrast, the fake piezoelectric performance would be always obtained when just using the apex or bottom of electric signals that contain the triboelectric part and piezoelectric part as the evaluation index. Therefore, our method CBA can be used for separating the piezoelectric component from the triboelectric-piezoelectric hybrid output, and evaluating piezoelectric performance without influenced by triboelectric signals.

## Discussion

In summary, this work elucidates the output signal composition of conventional piezoelectric devices, and develops an effective method to quantitatively extract the piezoelectric part from a hybrid output regardless of the interference from triboelectric signals caused by CE. For a typical sandwich-structured PVDF device, an SE-TENG and a PENG both exist during the compressing test, resulting in the triboelectric-piezoelectric hybrid output. Our method that analyzes the force signals of PVDF-based devices can differentiate the triboelectric part and piezoelectric part from the hybrid output in the time domain. More specifically, the triboelectric part occurs before and after the contact between the object and device, while the piezoelectric part only exists when the object and device are contacted. The piezoelectric charge transfer can be extracted from hybrid output using CBA method that analyzes the electrostatic balance among the Al plate, Kapton layer, electrodes, and PVDF film. Besides, the $d_{33}$ calculated by CBA is 33.74 pC/N, which is consistent with the results of OS method (33.20 pC/N) and DE method (32.64 pC/N). It is also verified by a commercial instrument, indicating that both of these three methods are effective. This research gives a deep insight on the output composition of piezoelectric devices in practical measurement and provides a simple method to separate the piezoelectric component from a hybrid output. It would help future researchers to evaluate true piezoelectric performance of flexible piezoelectric devices, which is beneficial for developing next-generation high-performance piezoelectric materials and promoting their application in wearable electronics.

## Methods

Fabrication of PVDF-based piezoelectric devices: A commercial poled PVDF film with silver electrode (110 μm, Measurement Specialties Inc.) was encapsulated by the Kapton tape (50 μm). The surface area of PVDF-based device was $3 \times 3$ cm$^2$.

Fabrication of PI-based non-piezoelectric devices: A polyimide (PI) film with a thickness of 125 μm was coated by silver paste on both two sides. Then, the Ag electrodes were obtained after sintering at 160 °C for 10 min. Finally, it was encapsulated using a Kapton tape (50 μm) for a PI-based non-piezoelectric device ($3 \times 3$ cm$^2$).

Measurements: The piezoelectric and non-piezoelectric devices were compressed repeatedly by a linear motor (LinMot E1100) to generate electric signals. An aluminum (Al) plate ($2 \times 2$ cm$^2$) fixed on the linear motor was used as an indenter to compress the PVDF-based and PI-based devices. Besides, fluorinated ethylene propylene (FEP) film and polyimide (PI) film were attached to the Al plate to compress these two kinds of devices. At the same time, the short-circuit current, open-circuit voltage and transferred charge were measured by an electrometer (6514, Keithley); the force data were obtained by using a force gauge (ZTS-DPU 100 N, IMADA) and the sampling rate was 2000 data points per second. The piezoelectric coefficient $d_{33}$ of the PVDF film was measured by a commercial instrument (ZMpiezo-P) provided by Foshan Zilm Technology Co. Ltd.

## Data availability

Source data are provided with this paper.

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

## Acknowledgements

We thank the National Natural Science Foundation of China (52061160482, 52005289), the Tsinghua University Spring Breeze Fund, the Local Innovative and Research Teams Project of Guangdong Pearl River Talents Program (2017BT01N111), Shenzhen Geim Graphene Center, Shenzhen Technical Project (JSGG20191129110201725), and the China Postdoctoral Science Foundation (Project no. 2020M670309).

## Author contributions

C.C. and C.Y. conceived the idea and designed the experiments. C.C. and S.Z. carried out the experiments and analyzed the results. C.P., Y.Z., and F.W. helped with the experiments. C.C. wrote the paper. All authors reviewed, discussed, and edited the manuscript. C.Y. and Z.W. supervised the project.

## Competing interests

The authors declare no competing interests.
