## [Peer Review File · Nature Communications]

REVIEWER COMMENTS

Reviewer #1 (Remarks to the Author):

The manuscript "A Method for Quantitatively Separating the Piezoelectric Component from the as-received "Piezoelectric" Signal" proposed an effective method for quantitatively identifying and extracting the piezoelectric signal from the triboelectric and piezoelectric hybrid signal using PVDF based piezoelectric device.

This study is expected to be of great help in effectively evaluating the performance of piezoelectric devices. It has sufficient novelty for the publication in Nature Communications. However, there also are several issues that the authors need to address carefully first

1. In this study, author utilized PVDF piezoelectric polymer for piezoelectric device. Is the proposed method universally applicable to other piezoelectric materials such as PZT, BTO, ZnO, or composite materials?
2. The author used Al plate as a pressing material. What about using other materials such as insulators?
3. Is there any difference in the tendency of piezoelectric signal and triboelectric signal according to applied pressure or frequency?
4. Are there any differences between piezoelectric signals and triboelectric signals in terms of shape?
5. Are there any factors we should consider when measuring the signal of a piezoelectric device? For example, thickness, elasticity, pressure, and etc.
6. "d33" should be modified to "effective d33".
7. There are many broken symbols in the main manuscript.

Reviewer #2 (Remarks to the Author):

The manuscript "A Method for Quantitatively Separating the Piezoelectric Component from the as-received "Piezoelectric" Signal" suggested a simple and effective method for quantitatively identifying and extracting the piezoelectric charge from the hybrid signal. The triboelectric and piezoelectric parts in the hybrid signal generated by a poly(vinylidene fluoride)-based device are clearly differentiated, and their force and charge characteristics in the time domain are identified. The authors described would help future researchers to evaluate true piezoelectric performance of flexible piezoelectric devices, which is beneficial for developing next-generation high performance

piezoelectric materials and promoting their application in wearable electronics. So I would recommend this manuscript for the possible publication in Nature Communications after Major revision.

Comment 1:

In Figure 3, the authors described the sole triboelectric signals are generated by the PI-based device. In the PI based device structure, the PI between the two electrodes could act as a capacitor to increase the capacitance and increase the output. I would like to ask if there is any other way to only analysis the triboelectric signal due to the charge transfer.

Comment 2:

In section 5, the authors described how to extract the piezoelectric charge transfer from “piezoelectric” output for piezoelectric performance evaluation. I recommend explaining the exact definition of piezoelectric charge transfer. If the authors want to describe the charge transfer by the piezoelectric effect, I recommend adding its charge transfer mechanism in detail.

Reviewer #3 (Remarks to the Author):

This manuscript describes the quantitative analysis of the output signal of the direct piezoelectric effect which include the triboelectric effect. Actually, many of studies have been reported in the field of piezoelectric and triboelectric energy harvesting, however, most of them include both effects and detailed quantitative analysis has scarcely been conducted because of the lack of the measurement methods which can separate each effect from the output signal.

In this study, authors provided the simple and precise measurement method which can extract the piezoelectric effect with eliminating the triboelectric effect of the piezoelectric materials. I think it is very useful not only to evaluate the output performance of the vibration or wearable energy harvesters but also to optimize the design of them. I have no major comments but minor comments are indicated below.

1. Polarization direction of Fig. 4 and Suppl. Fig. 6 is opposite. I think the measurement should use same PVDF films because Q_1 should be same. In this experiment, which method used: reversal of specimen or grand connection switch between front and back electrodes?
2. In page 15, DE and OS methods are indicated. Are they same as the measurements in Fig. 3?
3. How does output charge measured: charge amplifier or integral of current?

Point-by-point to response to reviewers' comments (comments in black, response in blue)

Reviewer #1 (Remarks to the Author):

The manuscript "A Method for Quantitatively Separating the Piezoelectric Component from the as-received "Piezoelectric" Signal" proposed an effective method for quantitatively identifying and extracting the piezoelectric signal from the triboelectric and piezoelectric hybrid signal using PVDF based piezoelectric device.

This study is expected to be of great help in effectively evaluating the performance of piezoelectric devices. It has sufficient novelty for the publication in Nature Communications. However, there also are several issues that the authors need to address carefully first.

Response: We sincerely thank the reviewer for his/her positive assessment for publishing our paper on Nature Communications.

1. In this study, author utilized PVDF piezoelectric polymer for piezoelectric device. Is the proposed method universally applicable to other piezoelectric materials such as PZT, BTO, ZnO, or composite materials?

Response: Thanks for your question. To answer this question, we have spent quite a long time collecting different kinds of samples for measurement. Although triboelectric signals and piezoelectric signals have distinctive differences in the time domain, the relative amplitude between them plays a crucial role for extracting the charge signals. Here we measured a commercial PZT device (650 pC/N) using the same experimental setup instead of the PVDF-based device. For PZT, as shown in Figure R1, the sole piezoelectric charge transfer in Fig. R1a and b is around 400 nC and -400 nC, respectively; this value is about 400 times larger than the triboelectric charge of the PVDF sample (about -1.0~1.5 nC) and thus the signal proportion from

CE can be ignored. Besides, the ideal coupling shape in the forward connection was not observed in Fig. R1c, while it was found in PVDF-based device (Fig. 4e). Therefore, we confirm that the triboelectric signals can be ignored when measuring those piezoelectric ceramics with high piezoelectric coefficient.

We also tried to measure the charge signals from ZnO single crystal film (~ 2.3 pC/N), but did not obtain significant result. Charge signals generated by ZnO film are small and they have the same level of noise in the measurement, and thus we cannot extract piezoelectric charges precisely. For instance, the piezoelectric charge only increased by 23 pC when increasing 10 N force in theory, which is about 500 times lower than triboelectric charge. Regrettably, the ZnO film that we obtained is a brittle matter that easily cracks under high force loading during our experiment (Figure R2), which also increases the difficulty to measure its piezoelectric signal. Based on existing data, it's reasonable to conclude that, our method is well applicable to piezoelectric polymers, e.g. PVDF and its co-polymer, but not very suitable to investigate those piezoelectric ceramics with high piezoelectric coefficient. Whether it is applicable to low piezoelectric materials needs to be further verified by improving the electric signal measurement condition, which is our next research topic.

Figure R1. Measuring charge signals generated by PZT device from 40.3 N to 57 N. The charge signals generated by shielded PZT device in (a) forward direction and (b) reverse direction. The charge signals generated by PZT device in (c) forward direction and (d) reverse direction; no coupled triboelectric-piezoelectric waveform is observed, indicating triboelectric charges can be ignored in this case.

Figure R2. The ZnO single crystal film cracked after testing.

2. The author used Al plate as a pressing material. What about using other materials such as insulators?

Response: Using different compressing materials instead of Al plate can definitely alter the triboelectric signals. To answer this question, we supplemented experiments using FEP, PI and Al plate to compress the PI-based device. The positive signals can be obtained when using FEP, while negative signals can be obtained when using Al plate. This is because, FEP can gain electrons from Kapton layer, while Kapton layer can gain electrons from Al plate. Besides, a small amount of triboelectric charge transfer also occurred between PI and Kapton layer. We added Supplementary Fig. 4 to demonstrate the influence of pressing materials. We also added the discussion in the main text, which is shown below:

“Furthermore, three compressing materials with different electron affinities were introduced including fluorinated ethylene propylene (FEP), PI, and Al, to compress the Kapton layer. Compared with Kapton, FEP tends to gain electrons, while Al tends to lose electrons. As a result, the directions of charge curves are opposite between FEP-Kapton and Al-Kapton (Supplementary Fig. 4 a, c). On the other hand, the PI-Kapton triboelectric series can also induce charge transfer (Supplementary Fig. 4b). Transferred charges obtained using different compressing materials are compared in Supplementary Fig. 4d. Therefore, the SE-TENG composed of compressing material and protective layer was confirmed by the experiments, which is consistent with the theoretical analysis.”

Supplementary Fig. 4 The influence of different compressing materials on triboelectric signal. The PI-based device is compressed by (a) FEP, (b) PI film, and (c) Al, resulting in different charge curves. (d) The transferred charge of FEP-Kapton is about 1 nC, that of PI-Kapton is ~ 0.3 nC, and that of Al-Kapton is -1.5 nC. Since these pressing materials have different electron affinities, the transferred charges have different directions and values.

3. Is there any difference in the tendency of piezoelectric signal and triboelectric signal according to applied pressure or frequency?

Response: We found that these two signals have different trends with increased force, while they both show the frequency-independent property at a certain force. This is because, with the increased force, the contact area between Al plate and device increased accordingly, resulting in signal increase; piezoelectric signals would also increase due to the intrinsic property of piezoelectric materials. In our experiment, piezoelectric signals show good linearity from 40 N to 90 N, and triboelectric signals show gradually saturated characteristics. Moreover, piezoelectric signals and triboelectric signals are unchanged with the increased frequency due to the fixed force and unchanged contact area. We added Supplementary. Figs. 7, 8 to demonstrate the influence from pressure and frequency, respectively, and added discussion in the main text:

“Triboelectric signals are mainly dependent on contact area, while piezoelectric signals are dependent on force. Consequently, they both increased from 40 N to 90 N due to the increased contact area and forces (Supplementary Fig. 7a, b), but the piezoelectric signals show good linearity compared with piezoelectric signals (Supplementary Fig. 7c). These two kinds of signals show frequency-independent behavior at low frequency from 0.5 Hz to 2.5 Hz (Supplementary Fig. 8a, b) due to the fixed force.”

Supplementary Fig. 7 Comparison of the trends of triboelectric signals and piezoelectric signals from 40 N to 90 N. (a) Triboelectric signals. (b) Piezoelectric signals. (c) Voltage-force relationships of triboelectric signals and piezoelectric signals.

Supplementary Fig. 8 The frequency influence on triboelectric signals and piezoelectric signals. (a) Triboelectric charge transfer and (b) piezoelectric charge transfer are obtained from 0.5 Hz to 2.5 Hz.

4. Are there any differences between piezoelectric signals and triboelectric signals in terms of shape?

Response: This is an interesting question. Output signals from PENG and TENG are related to the Maxwell's displacement current, indicating they are the functions of displacement; detailed information can be found in reference 26 in the manuscript. In our measurement, we found that triboelectric signals have longer response time (115.61 ms) than piezoelectric signals (32.81 ms), which is caused by the difference in displacement distance. We supplemented a discussion in the main text for better understanding their difference, as shown below. The revised Fig. 3 is also shown below.

“Comparing these two kinds of sole signals, the rising edge of the piezoelectric charge signal (Fig. 3h) is non-smooth that is consistent with the force signal (Fig. 3j); while triboelectric signals do not show this characteristic (Fig. 3e, i). Besides, triboelectric signals have a longer response time (115.61 ms) than piezoelectric signals (32.81 ms). This is because, although triboelectric signals and piezoelectric signals are the results of the displacement current, their actual displacement distances are different²⁶. Triboelectric signals can be generated over longer displacement (up to millimeter scale²⁷) because of the electrostatic induction, resulting in a longer response time. The limited deformation of PVDF film causes shorter displacement that enables piezoelectric signals to reach peak more quickly”

Fig. 3 Using loading force signals to differentiate the sole triboelectric signals and the sole piezoelectric signals. **a** Schematic diagram of measuring sole triboelectric signals from the PI-based device. Sole triboelectric signals including **c** short-circuit current, **e** open-circuit voltage, and **g** transferred charge are obtained under a certain **i** force signal, respectively. **b** Schematic diagram of measuring sole piezoelectric signals from PVDF-based device with a shielding layer. The conductive shielding layer is attached on the device's surface, which can guide the induced triboelectric charges (I_T) flow to the ground and only piezoelectric signals (I_P) can be obtained in the front electrode. Sole piezoelectric signals including **d** short-circuit current, **f** open-circuit voltage, and **h** transferred charge are obtained under a certain **j** force signal, respectively.

5. Are there any factors we should consider when measuring the signal of a piezoelectric device? For example, thickness, elasticity, pressure, and etc.

Response: Yes there are. A piezoelectric device with a larger thickness can produce larger voltage signal. For piezoelectric materials with lower elasticity, they can generate larger current signal. With the increased pressure, the signal output from piezoelectric devices would increase because of the piezoelectric effect. Besides, charge output of piezoelectric devices is determined by the type of piezoelectric materials. Reference 26 in the manuscript provides a quantitative calculation method. (doi: 10.1016/j.mattod.2016.12.001).

6. “d33” should be modified to “effective d33”.

Response: We have corrected it in the manuscript.

7. There are many broken symbols in the main manuscript.

Response: The broken symbols that you observed may be due to the improper conversion of the manuscript to the pdf file. We have double-checked the manuscript and updated version so that to eliminate any symbol display errors.

Reviewer #2 (Remarks to the Author):

The manuscript “A Method for Quantitatively Separating the Piezoelectric Component from the as-received "Piezoelectric" Signal” suggested a simple and effective method for quantitatively identifying and extracting the piezoelectric charge from the hybrid signal. The triboelectric and piezoelectric parts in the hybrid signal generated by a poly(vinylidene fluoride)-based device are clearly differentiated, and their force and charge characteristics in the time domain are identified. The authors described would help future researchers to evaluate true piezoelectric performance of flexible piezoelectric devices, which is beneficial for developing next-generation high performance piezoelectric materials and promoting their application in wearable electronics. So I would recommend this manuscript for the possible publication in Nature Communications after Major revision.

Response: We sincerely thank the reviewer for his/her positive comments.

Comment 1:

In Figure 3, the authors described the sole triboelectric signals are generated by the PI-based device. In the PI based device structure, the PI between the two electrodes could act as a capacitor to increase the capacitance and increase the output. I would like to ask if there is any other way to only analysis the triboelectric signal due to the charge transfer.

Response: Yes, there is. As shown in Supplementary Fig 3, we normalized the front electrode and back electrode into one electrode and removed the PI layer to demonstrate the working mechanism of SE-TENG. This is because, the SE-TENG always exists no matter what kind of electrode we connected. The true charge transfer caused by triboelectric effect between Al plate and Kapton layer can be obtained using the device shown in this figure.

Supplementary Fig. 3 The working mechanism of the CE-TENG of the PI-based device. (a) When the Al plate and the device are contacted with each other, the same amount of positive and negative charges can be induced in the surface of the Al plate and Kapton layer respectively. (b) When the Al plate is separating from the device, the positive charges in the Al plate flow to the electrode of the device. (c) When the Al plate is well separated with the device, there is an electrostatic balance between the Kapton layer and the electrode. (d) When the Al plate is contacting with the device, the positive charges in the electrode flow to the Al plate.

Comment 2:

In section 5, the authors described how to extract the piezoelectric charge transfer from “piezoelectric” output for piezoelectric performance evaluation. I recommend explaining the exact definition of piezoelectric charge transfer. If the authors want to describe the charge transfer by the piezoelectric effect, I recommend adding its charge transfer mechanism in detail.

Response: Thank the reviewer for this good suggestion. It is very important to give specific definition of the piezoelectric charge transfer to help future researchers evaluate piezoelectric materials performance. We added the specific definition of piezoelectric charge transfer Q_p in the manuscript and explained its working mechanism, as shown below.

“Here, we define Q_p as the piezoelectric charge transfer that induced by PVDF film at a certain force. Piezoelectric charges would accumulate with the increased force at the compressing stage and saturate in the compressed stage, finally get released in the releasing stage. During the above stages, the saturated charge value at the end point of compressing stage (start point of compressed stage) is Q_p .”

Reviewer #3 (Remarks to the Author):

This manuscript describes the quantitative analysis of the output signal of the direct piezoelectric effect which include the triboelectric effect. Actually, many of studies have been reported in the field of piezoelectric and triboelectric energy harvesting, however, most of them include both effects and detailed quantitative analysis has scarcely been conducted because of the lack of the measurement methods which can separate each effect from the output signal.

In this study, authors provided the simple and precise measurement method which can extract the piezoelectric effect with eliminating the triboelectric effect of the piezoelectric materials. I think it is very useful not only to evaluate the output performance of the vibration or wearable energy harvesters but also to optimize the design of them. I have no major comments but minor comments are indicated below.

Response: We highly appreciate the reviewer's comments regarding the novelty and influence, and share his/her view about our work that can help future researchers evaluate and optimize energy harvesters.

1. Polarization direction of Fig. 4 and Suppl. Fig. 6 is opposite. I think the measurement should use same PVDF films because Q1 should be same. In this experiment, which method used: reversal of specimen or grand connection switch between front and back electrodes?

Response: We thank the reviewer for this comment. Below, we address the reviewer's concerns point by point.

(1) The polarization direction of Fig. 4 and Suppl.Fig. 9 (rearranged) was set to be opposite. The triboelectric charges obtained from the front and back electrodes are not equal, which is illustrated in Fig. 3g. In other words, the triboelectric charges were almost induced in the front electrode, so we only measured the charge transfer in the

front electrode (back electrode was grounded) to keep the amount of triboelectric charge unchanged. For people to better understand our experimental set-up, we rearranged these two figures by adding the legends of Electrode 1 and Electrode 2. The revised figures are shown below.

Fig. 4 The triboelectric-piezoelectric hybrid output generated by the PVDF-based device with the negative polarization direction. **a** Schematic of measuring the triboelectric-piezoelectric hybrid output in the negative polarization direction of the PVDF film. **b** The signal generation process of hybrid output contains six stages including contacting, contacted, compressing, releasing, released, and separating stages. **c** The sole triboelectric signals (I_{sc} , V_{oc} , and charge) are generated by the PI-based device at 60 N. Sole triboelectric signals are generated in the stages of contacting and separating (light yellow area). **d** The sole piezoelectric signals (I_{sc} , V_{oc} , and charge) are generated by the PVDF-based device with shielding layer at 60 N. Sole piezoelectric signals are generated in the stages of compressing and releasing

(light green area). **e** The triboelectric-piezoelectric hybrid signals (I_{SC} , V_{OC} , and charge) are generated by the PVDF-based device at 60 N. The hybrid signals exist in the stages of contacting, compressing, releasing, and separating.

Supplementary Fig. 9 The triboelectric-piezoelectric hybrid output generated by the PVDF-based device with the positive polarization direction. (a) Schematic of measuring the triboelectric-piezoelectric hybrid output in the positive polarization direction of the PVDF film. (b) The signal generation process of hybrid output contains six stages including contacting, contacted, compressing, releasing, released, separating. (c) The sole triboelectric signals (I_{SC} , V_{OC} , and charge) are generated by the PI-based device at 60 N. Sole triboelectric signals are generated in the stages of contacting and separating (light yellow area). (d) The sole piezoelectric signals (I_{SC} ,

V_{OC} , and charge) are generated by the PVDF-based device with a shielding layer at 60 N. Sole piezoelectric signals are generated in the stages of compressing and releasing (light green area). (d) The triboelectric-piezoelectric hybrid signals (I_{SC} , V_{OC} , and charge) are generated by the PVDF-based device at 60 N. The hybrid signals exist in the stages of contacting, compressing, releasing, and separating.

(2) During our whole experiments, we used the same PVDF film to conduct all electrical measurements.

(3) We switched the ground connection between the front and back electrodes in Fig. 2 and 3, and the samples were fixed on the acrylic substrate. In sections 4 and 5, we used both the above two methods.

Please refer to the revised Fig. 4, Suppl. Fig. 9, and Fig. 5. Moreover, we also revised Fig. 5 for better understanding the experiment detail of extracting piezoelectric charge transfer.

Fig. 5 Evaluating the piezoelectric performance of PVDF film by extracting the piezoelectric charge transfer from the hybrid output. **a** The electrostatic balance is achieved when the Al plate and the PVDF-based device are contacted. I represents the positive polarization, II represents the negative polarization, Q_3 is the total charge in the Al plate, $-Q_2$ is the total charge in the Kapton layer, q is the charge transfer between electrodes in the positive polarization, Q_p is the induced piezoelectric charge, $-q'$ is the charge transfer between electrodes in the negative polarization. The transferred charges between electrodes are measured from the **b** positive and **c** negative polarization sides of PVDF-based device, and the applied force is 60 N. **d** The charge-force curves of PVDF-based device are obtained by three methods

including outside shielding (OS), compressed balance analysis (CBA), and directly extracting (DE). The effective d_{33} of OS is 33.20 pC/N, that of CBA is 33.74 pC/N, and that of DE is 32.64 pC/N. **e** The piezoelectric charge transfer under different forces is obtained by three methods including OS, CBA, and DE. They are almost equal.

2. In page 15, DE and OS methods are indicated. Are they same as the measurements in Fig. 3?

Response: Thank the reviewer for raising this valuable question. The OS method adapted the same measurement as shown in Fig. 3, while the measurement used in DE method was the same as are shown in Fig. 4 and 5. For the DE method, we flipped the device and measured the charge output only from the front electrode to obtain the same amount of triboelectric charges. Detailed information can refer to our answer (1) for your question #1.

3. How does output charge measured: charge amplifier or integral of current?

Response: A commercial electrometer (6514, Keithley) was used to measure the output charge.

REVIEWERS' COMMENTS

Reviewer #1 (Remarks to the Author):

The manuscript "A Method for Quantitatively Separating the Piezoelectric Component from the as-received "Piezoelectric" Signal" is well written, and it has sufficient novelty after revision. Therefore, I would recommend this manuscript for the possible publication in Nature Communications.

Reviewer #2 (Remarks to the Author):

Thank you for kindly responding to reviewer's comments. I think the authors reply to my question and refer to my opinion well. This manuscript has become more clear to understand. I would recommend this manuscript for the possible publication in Nature Communications.

Reviewer #3 (Remarks to the Author):

The authors have addressed my concerns satisfactorily and I think this manuscript can be accepted. Before publication, I suggest the authors to check the following minor comments.

1. "V" in Fig. 4a and Suppl. Fig. 9a is very confusing whether it is an electrometer same as Fig. 5 or external voltage which is applied for the poling treatment. I think it is an electrometer and if so please clarify that.

2. As reviewer #1 pointed out, there are still some garbled characters in my PDF, i.e. figure caption of Fig. 5. Please check the font of "II", "Q3" and the other symbols in the main text.

Point-by-point to response to reviewers' comments (comments in black, response in blue)

Reviewer #1 (Remarks to the Author):

The manuscript “A Method for Quantitatively Separating the Piezoelectric Component from the as-received "Piezoelectric" Signal” is well written, and it has sufficient novelty after revision. Therefore, I would recommend this manuscript for the possible publication in Nature Communications.

Response: We sincerely thank the reviewer for his/her positive comments and the approval on our work.

Reviewer #2 (Remarks to the Author):

Thank you for kindly responding to reviewer's comments. I think the authors reply to my question and refer to my opinion well. This manuscript has become more clear to understand. I would recommend this manuscript for the possible publication in Nature Communications.

Response: We appreciate the reviewer's positive comments and the responsible review of our work.

Reviewer #3 (Remarks to the Author):

The authors have addressed my concerns satisfactorily and I think this manuscript can be accepted. Before publication, I suggest the authors to check the following minor comments.

Response: We thank the reviewer a lot for the detailed comments.

1. "V" in Fig. 4a and Suppl. Fig. 9a is very confusing whether it is an electrometer same as Fig. 5 or external voltage which is applied for the poling treatment. I think it is an electrometer and if so please clarify that.

Response: Thanks for pointing out this issue. The symbol "V" represents the electrometer, we have revised Fig. 4a and Suppl. Fig. 9a. Also, the "V" in Fig. 1c is replaced by "R" for better understanding. All revised figures are shown below.

Fig. 4 The triboelectric-piezoelectric hybrid output generated by the PVDF-based device with the negative polarization direction. a Schematic of measuring the triboelectric-piezoelectric hybrid output in the negative polarization direction of the PVDF film. b The signal generation process of hybrid output contains six stages including contacting, contacted, compressing, releasing, released, and separating stages. c The sole triboelectric signals (I_{sc} , V_{oc} , and charge) are generated by the PI-based device at 60 N. Sole triboelectric signals are generated in the stages of contacting and separating (light yellow area). d The sole piezoelectric signals (I_{sc} , V_{oc} , and charge) are generated by the PVDF-based device with shielding layer at 60 N. Sole piezoelectric signals are generated in the stages of compressing and releasing (light green area). e The triboelectric-piezoelectric hybrid signals (I_{sc} , V_{oc} , and charge) are generated by the PVDF-based device at 60 N. The hybrid signals exist in the stages of contacting, compressing, releasing, and separating.

Supplementary Fig. 9 The triboelectric-piezoelectric hybrid output generated by the PVDF-based device with the positive polarization direction. (a) Schematic of measuring the triboelectric-piezoelectric hybrid output in the positive polarization direction of the PVDF film. (b) The signal generation process of hybrid output contains six stages including contacting, contacted, compressing, releasing, released, separating. (c) The sole triboelectric signals (I_{sc} , V_{oc} , and charge) are generated by the PI-based device at 60 N. Sole triboelectric signals are generated in the stages of contacting and separating (light yellow area). (d) The sole piezoelectric signals (I_{sc} , V_{oc} , and charge) are generated by the PVDF-based device with a shielding layer at 60 N. Sole piezoelectric signals are generated in the stages of compressing and releasing (light green area). (e) The triboelectric-piezoelectric hybrid signals (I_{sc} , V_{oc} , and charge) are generated by the PVDF-based device at 60 N. The hybrid

signals exist in the stages of contacting, compressing, releasing, and separating.

Fig. 1 Schematic of the existence of both SE-TENG and PENG during practical application of piezoelectric devices, but the time at which the signal is generated has distinct differences. a The human finger has positive charges and the surface of a piezoelectric pixel has negative charges (left). The piezoelectric material would experience elastic deformation when a human finger is pressing it and the piezoelectric effect would occur (right). b A SE-TENG is formed when using the positive finger touches the negative piezoelectric pixel again. The working mechanism of SE-TENG contains four stages including contacted, separating, separated, and contacting, the triboelectric signals will be generated in the stages of separating and contacting. c The working mechanism of PENG contains four stages including initial, compressing, compressed, and releasing. Piezoelectric signals will be generated in the stages of pressing and releasing.

2. As reviewr #1 pointed out, there are still some garbled characters in my PDF, i.e. figure caption of Fig. 5. Please check the font of "II", "Q3" and the other symbols in the main text.

Response: Thanks for your kind reminder. We have reinput the relevant variable symbols in the main text and double-checked the manuscript to eliminate the display errors